# Identifying central elements of the therapeutic alliance in the setting of telerehabilitation: A qualitative study

**Barbara Seebacher** [1,2]*, **Carole Geimer**[3], **Julia Neu**[3], **Maria Schwarz**[4], **Gudrun Diermayr**[3]

**1** Clinical Department of Neurology, Medical University of Innsbruck, Innsbruck, Austria, **2** Department of Rehabilitation Science, Clinic for Rehabilitation Münster, Münster, Austria, **3** School of Therapeutic Sciences, SRH University Heidelberg, Heidelberg, Germany, **4** Department of Psychosocial Rehabilitation, Clinic for Rehabilitation Münster, Münster, Österreich

* barbara.seebacher@i-med.ac.at

## Abstract

### Introduction

Therapeutic alliance is a relevant aspect of healthcare and may influence patient outcomes. So far, little is known about the therapeutic alliance in telerehabilitation.

### Purpose

To identify and describe central elements of therapeutic alliance in the setting of telerehabilitation and compare it to those in conventional rehabilitation.

### Methods

In this qualitative study, a literature search and in-depth semi-structured interviews with rehabilitation and telerehabilitation experts were conducted from 15.5.-10.8.2020 on elements influencing the therapeutic alliance in rehabilitation and telerehabilitation. Using a combined deductive and inductive approach, qualitative content analysis was used to identify categories and derive central themes.

### Results

The elements bond, communication, agreement on goals and tasks and external factors were identified in the literature search and informed the development of the interview guide. Twelve purposively sampled experts from the fields of physiotherapy, occupational therapy, speech and language therapy, psychology, general medicine, sports science and telerehabilitation software development participated in the interviews. We identified three central themes: building effective communication; nurturing a mutual relationship of trust and respect; and agreement on goals and tasks and drivers of motivation.

### Conclusions

In this qualitative study, key elements of therapeutic alliance in rehabilitation confirmed those reported in the literature, with additional elements in telerehabilitation comprising

**Data Availability Statement:** All relevant data are within the manuscript and its Supporting information files.

**Funding:** The authors received no specific funding for this work.

**Competing interests:** The authors have declared that no competing interests exist.

support from others for ensuring physical safety and technical connectedness, caregivers acting as co-therapists and applying professional touch, and promoting patient autonomy and motivation using specific strategies.

## Introduction

Recent advances in the development of new technologies have pushed telerehabilitation as an add-on intervention to conventional rehabilitation. Telerehabilitation is the provision of remote rehabilitative services using information and communication technologies [1]. Telerehabilitation can reduce costs and facilitate the geographical accessibility of rehabilitation facilities in rural areas [1]. Furthermore, telerehabilitation allows for improved treatment continuity, an increase in the frequency of therapy [2], and promotes independence and self-efficacy in the home environment [1].

Despite this added value of telerehabilitation, technical problems, an inadequate internet connection or a low level of technical know-how of the therapist or patient impede its widespread application [2]. In addition, hands-on therapies cannot be carried out using telerehabilitation due to the lack of the therapist's physical presence [3], and non-verbal messages are more difficult to interpret [4]. The latter aspects play a central role in motor rehabilitation in physiotherapy, occupational therapy, and speech and language therapy and can possibly influence the interaction between therapist and patient (i.e., the therapeutic alliance).

Therapeutic alliance (TA), which originated in psychotherapy [5], has been defined as 'a mutual collaboration and partnership between the therapist and client' [6, page 1]. Central elements of TA are an individualised treatment, respect for human dignity and rights, congruence [7] and empathy [8]. Other aspects include a successful communication [9] and the promotion of self-efficacy and autonomy of those being treated [10].

Different concepts of TA have been proposed in the literature [11–13]. The well-known TA model of Bordin encompasses the agreement between the patient and therapist on the goals of therapy, the agreement on tasks to be worked on during therapy, and the quality of the emotional connection ('bond') including aspects of trust, respect and caring [11]. This alliance thus describes the quality of the working relationship between patient and therapist [11]. Recent meta-analyses have shown that successful TA is a significant predictor of reduced symptoms and a positive psychotherapy outcome [6, 10, 14].

Within the last decade, the term TA has increasingly been expanded to the context of social and health professions such as nursing, social work, counseling, psychiatry, rehabilitation [13] and mental e-health [15]. Moreover, TA has been explored in physiotherapy [16–19], occupational [20] and speech and language therapy [21]. Among others 'stimulus to activity' [20], 'contextual shapers', e.g., the patient's family [21] and 'establishing connections with the body being central' [16] emerged as relevant themes of TA within motor rehabilitation.

TA in telerehabilitation requires special attention due to several key factors. Firstly, utilising technology-mediated communication in telerehabilitation associated with the potential disruptions from technical issues may affect the establishment and maintenance of the TA. Secondly, telerehabilitation may complicate deciphering non-verbal cues, which hold a significant role in face-to-face interactions and can contribute to the development of TA. Finally, the challenges of building trust and sustaining patient engagement may be heightened in telerehabilitation due to the constraints of limited physical presence. Although existing frameworks of TA have been adapted to the fields of physio-, occupational or speech and language therapy,

little is known about TA in telerehabilitation in the respective fields. Therefore, this study aimed to identify and describe the central elements of TA in the setting of telerehabilitation as experienced by physiotherapists, occupational and speech and language therapists and thereby expand those of conventional rehabilitation and compare it to those in conventional rehabilitation.

## Materials and methods

### Theoretical framework

By addressing the relationship between the patient and therapist as viewed from different experts' perspectives, a social constructivist worldview or research paradigm was chosen for this study [22]. Researchers recognised that both their interviewees and they themselves perceive and understand reality in a different way, embracing the ontological assumption of multiple truths and multiple realities [23]. They strived to uncover the implied meaning or latent content in interviewees' experiences, reflected by themes [24]. Moreover, researchers acknowledge that their personal, professional, historical, and cultural backgrounds and experiences (S1 Table) influenced their interpretation of results [22].

Structuring qualitative content analysis according to Steigleder (2008) [25] appeared particularly suitable for the inductive approach to the expert interview analysis due to its focus on the methodically controlled, empirically guided revision of category systems.

### Research design, ethics, and data protection

To gain insight into the TA in the setting of telerehabilitation, a combined deductive and inductive qualitative approach was chosen. Based on the results from a literature search, salient elements of the TA were identified from which interview guides were developed individually for each interviewee, depending on their professional background. The methodological approach and reporting were guided by the 'Consolidated Criteria for Reporting Qualitative Research' (COREQ) [26] (S1 File). The active study duration was from 15.5.2020 to 10.8.2020, followed by the data analysis.

This study did not fall under the responsibility of the research ethics committees of the Medical University of Innsbruck, Austria, and SRH University Heidelberg, Germany as only academic, telerehabilitation and clinical experts were involved. The study was conducted according to the European Union General Data Protection Regulation (DSGVO 20216/679), the Austrian (DSG 2019) and German (BDSG 2017) Data Protection Laws, ethical principles stated in the Declaration of Helsinki (2013) and prospectively registered as part of a questionnaire development study in the ISRCTN Registry (ISRCTN10132326) on the 2.5.2020. Participant data were pseudonymised using an ID number and a separately stored coding list. Before the interviews began, all experts received a detailed letter explaining the research project and gave their written consent for participation and recording the interview. The participant identification list is only accessible to one person in the research group and will be destroyed after 10 years. Until then, the data is kept on a password-protected computer in an encrypted file.

### Literature search

A literature search was conducted in April 2020 in the databases PubMed and Cochrane Library as well as in Google Scholar using the keywords detailed in the S2 Table. Aims of the literature review were to complement Bordin's theoretical construct of TA within rehabilitation and telerehabilitation, particularly within the domains of physiotherapy, occupational therapy, or speech and language therapy. Peer-reviewed full-text articles, books, and

conference proceedings in English or German language were included, that investigated relevant aspects of TA within rehabilitation or telerehabilitation and identified gaps in the literature, explored new perspectives, or proposed enhancements to existing TA frameworks. Based on the literature search results and researchers' clinical expertise the interview guides were designed.

### Participants and recruitment

Participants were recruited between 15.5.2020 and 10.8.2020. Experts were accessed via telephone or email by the author who conducted the qualitative interviews (CG), who had access to information that could identify individual participants during data collection. Semi-structured interviews with experts from rehabilitation and tele-rehabilitation on relevant elements influencing successful TA were conducted. Experts are understood as persons who have special knowledge due to their professional position or training [27]. The sample size was set in advance at 8–12 experts from different fields [28]. These were categorised into the following groups, each requiring a minimum of 2 experts, and purposive quota sampling [29] was performed: (1) experienced therapists in the field of physiotherapy, occupational therapy, and speech and language therapy; (2) therapists with an additional degree in psychology, (3) therapists with experience in using telerehabilitation and (4) experts with technical knowledge in telerehabilitation and software development.

### Interview guides and interview procedures

The literature search results and clinical expertise of the researchers informed the development of the interview guides. Due to the different areas of knowledge and positions of the experts, the guides were designed individually for each expert group and adapted specifically for each expert (see S2 Table for elements of the literature search and an exemplary interview guide). In-depth semi-structured interviews included open questions about TA in motor rehabilitation in general [30–32], telerehabilitation and topics such as communication [33, 34], trust [35, 36], congruence [37], self-disclosure [37], respect [38], empathy [37, 39], therapist-patient relationship [37, 38], working alliance (incl. goal setting, tasks, motivation) [40–42], external influencing factors (e.g., relatives and/or caregivers, information and communication technology, safety, time) [21, 43, 44], roles and responsibilities [38, 42]. The interviews, which lasted 45–60 minutes, were conducted by telephone by a trained physiotherapist in the Master's programme (CG) and recorded using a Philips voice tracer. One interview was carried out with each expert and any open questions clarified during the interview. To establish a relationship with the interviewees, they were introduced to the researcher's professional background, personal goals, interests in the research topic and study purpose. Field notes were taken.

### Data analysis

Recordings were transcribed using the transcription software f4transcript and using semantic-content transcription rules according to Dresing and Pehl (2018) [45]. The subsequent data analysis was carried out using MaxQDA. A combined deductive and inductive qualitative approach was chosen. The theoretical framework of TA as derived from the literature search was further used to determine the initial main categories (pre-coding), whereas the refined main categories and sub-categories were created inductively from the interview material. To condense the data material to content of particular importance for the research project, the Steigleder (2008) modified variant of a structuring qualitative content analysis was used [25]. A latent qualitative content analysis approach was used to be able to interpret the underlying meaning of the text [24]. Representative quotations were assigned to main and sub-categories.

The initial coding steps were performed by the second (CG) and third authors (JN) who double-coded all material [46]. Ensuring reliability is crucial in content analysis [47] because it guarantees consistency in coding decisions and assesses the rigour and transparency of the coding frame and its application to the data [48, 49]. To determine intercoder reliability, Cohen's kappa along with its 95% confidence interval (CI) was calculated [50, 51] using GraphPad Prism 9, San Diego, California. McHugh recommends interpreting kappa values as follows: 0–0.20 indicating no agreement, 0.21–0.39 minimum agreement, 0.40–0.59 weak agreement, 0.60–0.79 moderate agreement, 0.80–0.90 strong agreement, and 0.91–1 almost perfect agreement [52].

Peer scrutiny and debriefing among the researchers were used throughout the process. Using a sequential process, the coding frame was constantly evaluated and adapted to the interview material [25]. Analysis steps involved familiarisation with the entire text material (preparation, step 1); structuring the material marking relevant text segments, establishing a criterion of segmentation and identifying units of analysis (pre-structuring, step 2); developing the main categories from the literature and theory of TA, based on findings from the text and the research question (pre-coding, step 3); specification, compilation, evaluation and revision of the subcategories and creation of a coding frame (step 4); specification of the coding rules including definitions, examination, and revision of categories (step 5). Further steps included screening of the material, coding frame-based grouping of relevant categories and iterative revisions of the coding frame; marking and saving of not assignable text segments as an external data file (step 6); examining the allocation of the text segments to the most appropriate subcategories (step 7); revising and extracting relevant text segments into a table; checking of the categories and extracted text segments for plausibility and congruency in content and creating central themes (step 8); and describing the results (step 9) [25].

## Results

### Literature search

A total of 7,933 articles were initially identified after removing duplicates. Upon screening titles and abstracts, 7,821 studies were excluded. Subsequently, 182 potentially relevant papers underwent full-text evaluation. Following the full-text screening process, 145 studies were excluded for the following reasons: the evaluation solely focused on the effects of interventions on TA (n = 38); exclusive focus on the perceptions of patients or therapists regarding TA (n = 15); utilisation of Bordin's theoretical construct of TA without extension (n = 14); investigation of TA in the domain of psychotherapy or psychology (n = 57); or exploration of TA in the field of telemedicine or digital healthcare (n = 23). Finally, 35 articles were included.

Based on the construct of TA by Bordin (1979) [11] and the results of the literature review, elements and issues influencing successful TA in rehabilitation were identified, which informed the interview guide development: relationship between the patient and therapist, trust, communication, agreement on goals and tasks and external influencing factors. TA was explored in diverse contexts, contributing valuable insights into its dynamics.

**Relationship between the patient and therapist.** The essence of the TA lies in the intricate bond between the patient and therapist, encompassing personal, emotional, and professional dimensions. The therapist's crucial role involves displaying empathy and responsiveness, fostering mutual sharing of emotions, and engaging in self-revelation to build a strong personal relationship [37]. Emphasising the importance of a common focus among all parties, the patient should be forefronted, transcending the focus from the illness to their individual physical, psychological, cultural, and social characteristics and needs in relation to personal goals and therapy contents [30, 36, 38, 53]. Establishing a profound emotional and

professional connection is essential for effective knowledge transmission at a professional level [40].

To maintain a client-centered approach, a clear distribution of roles and responsibilities is imperative to prevent conflict and encourage the active participation of the patient in the therapeutic process [41, 42]. Shared responsibility, identified as a key factor [32], necessitates a delicate balance between patient autonomy and therapist support, presenting a significant challenge [54]. Introducing humor serves to relegate the illness to the background, fostering an improved mood [37]. The therapist's congruence, involving genuine, open, and authentic interactions, is pivotal at a professional level [37]. The physical presence of the therapist is deemed relevant, especially in tele-rehabilitation, where ensuring safety [55], conducting a comprehensive physical examination [55, 56], and providing both psychological and physical support are challenging [31, 32]. Lack of physical presence emerges as a barrier to TA in tele-rehabilitation [44].

**Trust.** Various aspects contribute to building mutual trust between the patient and therapist, including the exchange of information [38, 57], positive feedback, empathy and respect [38], along with sense of safety [55]. In tele-rehabilitation, the absence of direct safety guarantees poses a challenge to trust [35]. Information exchange is foundational for TA development [32], impacting interaction, satisfaction, and therapy success [58].

**Communication.** Effective communication enhances the rehabilitation process, enabling mutual understanding, shared decision-making, and person-centered communication [59]. Different communication channels, such as nonverbal, verbal, and paraverbal communication play crucial roles [18, 60, 61], with face-to-face communication being emphasised for nonverbal aspects [61]. Technical aspects in tele-rehabilitation [34], including distance [62] and time-delay [33, 63], technical aspects [34], limit non-verbal communication, underscoring the importance of face-to-face interaction [33].

**Agreement on goals and tasks.** Agreement on goals between the therapist and patient is pivotal for goal setting, decision-making [41], TA development, and therapy implementation [18]. Goal setting, aligned with patient needs, increases motivation [39], while the patient's expectation of success, achievement of goals [40], autonomy, and self-management contribute to long-term therapy success [64]. Patient engagement becomes especially crucial in tele-rehabilitation [65–67].

**External influencing factors.** Relatives and/or caregivers form an integral part of the TA, often referred to as a triad [21]. Their support plays a significant role in promoting the rehabilitation process [37]. In cases involving children, a shared relationship is observed between the therapist, child, and parents [38]. Relatives and/or caregivers provide technical support during therapy preparation and procedure, ensuring safety [42] and the surroundings can influence TA development [21, 44]. Time is recognised as a valuable resource, influencing trust-building [68], communication, and the consideration of patient needs [43]. In tele-rehabilitation, spending more time with the patient is highlighted as a factor promoting TA [55].

Main and secondary elements of TA according to the literature search are presented in S3 Table.

## Expert interviews

As shown in Table 1, 12 experts with different areas of expertise were interviewed in this phase. All experts approached by the research team agreed to being interviewed. Therapists worked in the fields of pediatrics, neurology, geriatrics, oncology, and musculoskeletal rehabilitation. Based on the results of the expert interviews, the elements already identified through the literature review were either confirmed, explored in more depth or new aspects were

**Table 1. Role and number of experts.**

| Field of professional experience and knowledge | Number and gender |
|---|---|
| Clinically experienced therapists with a MSc or PhD degree and some experience in telerehabilitation (physiotherapy, speech and language therapy, occupational therapy) | 3 female |
| Therapists with a MSc or PhD degree and an additional degree in psychology | 2 female |
| Therapists with a MSc degree and extensive experience in telerehabilitation | 2 female, 2 male |
| Experts with medical and technical knowledge with a PhD degree (general medicine, sports science and telerehabilitation software development) | 1 female, 2 male |

generated. Certain elements became apparent that are significant for a sustainable TA in telerehabilitation.

## Coding tree and themes

We identified 9 unique codes and 50 subcodes as potential elements related to TA through qualitative content analysis in our interview data (Table 2; see S4 Table for the full coding tree including code descriptors). The intercoder reliability for the raters was determined to be Kappa = 0.900 ($p<0.0001$), 95% CI (0.873, 0.927). Further synthesis generated three central themes: building effective communication (theme 1); nurturing a mutual relationship of trust and respect (theme 2); agreement on goals and tasks and drivers of motivation (theme 3). Representative quotes are presented with the themes.

We present these themes as a framework for potential facilitators associated with TA in telerehabilitation.

**Theme 1. Building effective communication.** Communication through words, facial expressions, gestures, and therapeutic touch is considered a crucial element of the TA in rehabilitation, whereas in telerehabilitation the technical connection needs to be established and maintained for communication to be uninterrupted, and touch can only be applied by others acting as co-therapists.

Healthcare experts considered both verbal and non-verbal communication an essential component of successful TA, with touch, so-called 'hands-on', being an essential source of the relationship between the patient and therapist.

'What also plays a big role, of course, is touch, I have had the experience that touch plays a very big role in therapy.'

(ID 11, section 41)

In telerehabilitation, the physical distance creates challenging situations for therapists, especially in situations where manual control or assistance is required.

'I felt that the patients felt well looked after and also that I demonstrated general exercises, but anything that would have needed hands-on somehow or a check with my hands, that didn't work out so well.'

(ID 11, section 7)

Moreover, facial expressions and gestures were mentioned as important elements of non-verbal communication in telerehabilitation. They would be of central importance when it

**Table 2. Codes derived from expert interview data.**

| Codes | Subcodes |
|---|---|
| **Communication** | Effective communication |
| | Clear instructions |
| | Mutual feedback |
| | Touch |
| | Advising on safety |
| | Appropriate means of communication |
| **Bond** | Safeguarding |
| | Presence |
| | Responsiveness |
| | Openness |
| | Self-disclosure |
| | Genuinely caring |
| | Respect |
| | Acceptance |
| | Honesty |
| | Friendly interactions |
| | Self-reflexion |
| | Congruence and authenticity |
| | Humour |
| | Appreciation |
| | Building an emotional relationship |
| | Building a professional relationship |
| **Trust** | Safety |
| | Respect |
| | Transparency |
| | Empathy |
| | Mutual exchange |
| | Continuous care |
| **Agreeing on goals and tasks** | Similarity of goals |
| | Shared responsibility |
| | Goal setting |
| | Defining tasks |
| | Targeting |
| | Individualising |
| | Encouraging |
| | Adherence |
| | Congruence |
| **Patient autonomy and self-management** | Being prepared |
| | Self-efficacy |
| | Self-management |
| | Autonomy |
| **Motivation of the patient** | Motivating factors in rehabilitation and telerehabilitation |
| | Further motivating factors |
| **Agreeing on roles** | Roles of the therapist |
| | Roles of the patient |
| | Roles of the family and caregivers |

(*Continued*)

**Table 2.** (Continued)

| Codes | Subcodes |
| --- | --- |
| **External factors** | Home environment |
| | Time |
| | Supporters |
| | Technology-related aspects to consider in telerehabilitation |
| | Potentially disruptive factors in telerehabilitation |
| **Differences between telerehabilitation and conventional therapy** | Preparation time |
| | Active role of the patient |
| | Change in interaction |
| | Hands-on |
| | Field of action |

came to assessing, whether the patient could adequately follow the instructions or how she/he was coping with the exercises. Video calls would be important in this regard.

'The advantage of the video is that you can really see what's between the lines, for example when the patient has a face contorted with pain or when she/he seems desperate.'

(ID 6, section 23)

However, the technical aspects of tele-rehabilitation would lead to restrictions in non-verbal communication, for example due to a limited field of vision caused by the positioning of the camera or poor image quality. Especially people with low affinity to technology would experience such problems.

'What was also difficult indeed were partly the lighting conditions. Not in our therapy practice, but with the elderly it is just real /. I don't know why they always have such dark apartments, but it was very dark, and you couldn't see the people very well, which makes it even more difficult to understand each other.'

(ID 8, section 11)

Experts also attributed a key role to effective verbal communication throughout the therapeutic process. In the process of goal setting, it would also be relevant for both the patient and the therapist to effectively exchange information and have open communication.

'When it comes to agreeing on goals, self-disclosure is of course very important, so that you actually get to know what the patient wants to achieve.' (ID 4, section 23). For the therapist, on the other hand, it would be essential from a professional point of view to disclose all necessary information to the patient that could influence her/his decisions and plans.

'Everything should be openly stated that could or should influence the patient's decision.'

(ID 5, section 9)

Even verbal communication can be negatively influenced by using technical aids due to background noise or a poor internet connection. Due to potential restrictions in verbal and non-verbal communication, the experts emphasised the importance of verbal guidance in

telerehabilitation. Different approaches were mentioned to ensure the understanding of the patients and getting around technological problems. This could be improved by simplifying verbal statements and by their precision and clarity.

'So, I have to guide a lot more verbally, which otherwise would not have been an issue non-verbally'.

(ID 1, section 21)

'I try to make my explanations even simpler'.

(ID 11, section 13)

If the patient had little technical 'know-how', verbal explanations would also be needed to make telerehabilitation possible at all, by supporting the implementation.

'And there is just, as I said, the communication with the elderly. Then the camera is sometimes askew, then somehow a woman in the background talks and there were already very many disruptive factors'.

(ID 8, section 11)

**Theme 2. Nurturing a mutual relationship of trust and respect.** An emotional and professional relationship between the patient and therapist is essential in rehabilitation, with the therapist being genuine, open, authentic, empathetic, humorous, and genuinely caring for the patient. Physical safety of the patient is a prerequisite for building trust and needs to be ensured in telerehabilitation by providing effective information and involving therapist-informed caregivers for technical and hands-on support. Additionally, professional competence of the therapist, his/her preparedness for telerehabilitation sessions, and mutual feedback in real-time are sources of TA in telerehabilitation. Clear, concise and individualised instructions and information delivered to the patient or caregivers to ensure their preparedness enhance understanding, professional relationship and trust.

Establishing a mutual understanding of trust and respect between the patient and her/his therapist was considered of central importance. The therapist should be genuine, open, authentic, and empathetic and should use humor in an individualised way to distract the patient from her/his illness.

'Trust, self-disclosure, empathy, and humor are equally important BUT the difference is in the approach to each patient. So, the way I deal with a patient humorously, or how I am empathic, or how I disclose my thoughts or feelings /. I think that needs to be individualised to each patient, although all aspects are equally important with each patient.'

(ID7, section 22)

Most important, the therapist should know the patient from face-to-face treatments before using telerehabilitation. She/he should be genuinely caring for the patient.

'The therapist must have experience with telerehab, at least in the use of a specific tool. But he must also know the patient / well, I'm not necessarily a friend of the therapist treating a patient he has never even seen, and that is unlikely to happen in practice. So, you have to know the patient and be caring for the patient.'

(ID 6, section 31)

In telerehabilitation, due to limited non-verbal communication or technical influences, mutual feedback would be particularly important to ensure that instructions or exercises were understood correctly. Both text messages and video calls could be used for this purpose. Direct and real-time feedback regarding the performance of therapeutic actions would be clearly preferable for the patient's learning process, as the following quote illustrates: 'Feedback, on whether an exercise is right or wrong, must always be in real time in order to learn from it'. (ID 9, section 25)

To facilitate the relationship, regular feedback on the course of therapy should be given from the therapist's and patient's perspective. Sufficient time resources should be allowed for this: 'Feedback should be given to avoid misunderstandings. So, it is always important to have the opportunity to clarify things calmly, that the communication is indeed cross-checked. So to speak, in fact you have to examine yourself whether what was understood is what you intended to say.' (ID 5, section 68)

The interview analysis showed that the experts consider trust as an essential component of TA. Trust, as part of the emotional relationship, would be influenced, among other things, by the exchange of information. The more communication there was, whether verbal or non-verbal, the greater the trust in the other person and thus the quality of the relationship.

'The more the patient knows what the therapist wants from her/him, and the therapist knows what the patient wants from her/him, the more trust can be built up, which is for the therapeutic alliance is strengthening'.

(ID 7, section 3).

Furthermore, safety plays an important role in building trust and further influences the patient's motivation to participate in therapy. According to the experts, the therapist could not motivate the patient without his or her trust, which would consequently lead to the patient dropping out of therapy.

'If the patient does not feel safe with me, either psychologically or physically,. . . then he loses his confidence, and I cannot motivate him,. . .'

(ID 10, section 67)

Trust in the therapist can thus be regarded as important for extrinsic motivation. As in conventional therapy, therapist's professional competence and preparation of the session is necessary in telerehabilitation. However, this differs significantly in terms of its scope. Telerehabilitation sessions must be planned precisely in advance and the patients must be informed about and provided with the necessary materials.

'The family can be of great importance in telerehab, especially with stroke patients, if the person can no longer use the computer, then family members have to set up the devices, to use the devices for example, or then also operate the computer.'

(ID 3, section 61)

**Theme 3. Agreement on goals and tasks and drivers of motivation.** Agreeing on goals and tasks is a relevant aspect of TA in rehabilitation, which can be implemented in

telerehabilitation based on patient autonomy and self-efficacy. Cognitive, physical, and technical support from others who may act as co-therapists may be necessary. Furthermore, shared decision-making, positive encouragement, and individualisation reflect well-functioning TA and together with goal attainment increase patients' autonomy and motivation for rehabilitation. Additional technical options are available in telerehabilitation such as reminders, reward systems and display of therapy success.

Setting goals tailored to the patient, continuity, and attaining goals would be relevant aspects through which trust between the patient and therapist could be built, and thus the success of the therapy enhanced.

'I try to build up trust with the patient with respect to the agreed goals and maintain a certain thread, along which the patient and I can navigate during the treatments.'

(ID 3, section 7)

'It is important responding to the wishes of the patient, recognising their goals, aiming to really work on the goal of the patient. That you give him feedback, that you say ok, this is your current status, that you could achieve with that effort, that feedback is also given regularly, so to speak, or a repeat test is carried out.'

(ID 2, section 18)

According to the experts, family members should possibly be involved in clarifying individual needs in the goal-setting process. This would apply, for example, in the case of impaired communication skills.

'I just had to do more parent counseling to query in which areas what is developing and then adjust together with the patients' caregivers whether we are now pursuing this goal or another goal.'

(ID 1, section 33)

Motivation could be influenced by additional aspects, for some of which relatives/caregiver support was necessary: the consideration of individual patient's needs and goal setting process, positive encouragement, facilitating patients' autonomy and self-efficacy, excellent preparedness for telerehabilitation, successful implementation of telerehabilitation and goal attainment. Goals should be realistic, achievable and time-limited. Regular adjustments of the therapeutic intervention to the patient's performance would increase motivation.

'You can motivate through needs, that you first find out what the basic need is, and that you hold out the prospect of fulfilling the need.'

(ID 4, section 71)

'I really try to think of the motivation and give the patient always a new input.'

(ID 10, section 59).

In telerehabilitation there would be additional possibilities to increase the motivation of the patients. These included reminders and reward systems (e.g., a high score) as well as displays of therapy progress. The experts reported a consistently high intrinsic motivation of the patients to participate in telerehabilitation. This was shown in different situations where no conventional therapy could take place.

'The fact that they did not have to be not afraid of (COVID-19) infection and could still do therapy, that they had the feeling that they can now use this time when they are home-bound. They were a bit worried because we often work very close, that they would catch it. They were fully motivated, that was great.'

(ID 1, section 69)

'In acute conditions, when a patient says that he fell and because of that he can't come to the therapy and asks: can we find an alternative?'

(ID11, sections 33 and 35)

Experts unanimously rated self-management and thus a high autonomy of the patients in conventional therapy as essential.

'So, for me, it's actually always the case that the responsibility, the recovery, always grows with the patient. So, he should make an effort, he should endeavor, and I support him with my expert knowledge.'

(ID 4, section 55)

They also referred to patients who consider the therapist responsible for the therapy and do not want to take personal responsibility.

'Some people, when they enter the therapy practice, they hand over self-responsibility at the door and don't take it back until they leave the room at the earliest.'

(ID 9, section 71)

Telerehabilitation would only succeed through the autonomous action of the person being treated or supportive persons, both preparing for and during the therapy session. Despite the possibility of verbal therapeutic support, the technical requirements for telerehabilitation have to be met and the necessary materials be prepared.

'They have to manage themselves more because there is just no-one facing them and putting anything (material) down to them directly, which means they already have to structure themselves.'

(ID 8, section 52)

'They have to be able to ensure their own safety, which means I can say, please see that just the table is nearby, put a chair to the side; or that the bed is nearby where they could then possibly sit down. I can give certain instructions or make suggestions, but the patient has to ensure that that's the way it is then.'

(ID 11, section 31).

The expert interviews revealed that relatives and/or caregivers are occasionally involved in the conventional therapy process, but even more so in telerehabilitation. Some patients have limited technical knowledge, or their impairments hinder the independent use of technical devices.

'That, for example, their daughter or niece or granddaughter supports them in using technology.'

(ID 11, section 35)

'Especially with stroke patients, if the person can no longer use the computer, then family members,. . . then they have to operate the computer.'

(ID 3, section 61)

Due to the physical distance in telerehabilitation, the therapist may not be able to support the patient to the necessary extent. Relatives and/or caregivers would therefore 'act as co-therapist' and as an 'extended arm of the therapist'. They would take over therapeutic tasks, such as palpation and manual assistance with exercises.

'That the mother, with smaller children, actually became the co-therapist. She currently has to do a lot of things that I would otherwise have done in therapy. And with neurological patients, it is also the case that the relatives have been instructed to do more co-therapy than they previously had to do.'

(ID1, section 39)

'If relatives are present, to involve them, e.g., that they "feel".'

(ID 11, section 13)

Relatives and/or caregivers could also ensure the safety of patients in telerehabilitation, as well as provide assistance with and support in carrying out a task in therapy.

'If a patient is at risk of falling, then I MUST ensure that a relative is present.'

(ID 11, section 15)

'I always told the parents to sit next to them so that they could help the child. I mailed or gave them sheets of paper, which they should then present to the child.'

(ID 1, section 11)

## Discussion

Using a literature search and semi-structured qualitative expert interviews, this study aimed to identify and describe central elements of TA in the setting of telerehabilitation and compare it to those in conventional rehabilitation. We identified three central themes from qualitative content analysis of interview data comprising building effective communication; nurturing a mutual relationship of trust and respect; and agreement on goals and tasks and drivers of motivation.

Most of the results of this study are consistent with those from telepsychology. According to Bischoff et al. (2004), therapists and patients need to be able to make necessary adjustments to communication patterns in order to balance the influence of technology and thus develop good TA [69]. This includes, for example, validating the patient's statements through extensive questioning by the therapist. Ghosh, McLaren, and Watson (1997) have found changes in communication in the context of telepsychology, such as the use of shorter sentences [70].

Simpson and Reid (2014) have described exchanges via a video phone call in telepsychology as promoting the development of patients' emotions and self-awareness as well as strong TA [71]. A review by Berger (2017) highlighted that active participation and assumption of responsibility by the client is significantly greater in internet-based psychotherapy [15]. This is in line with the relevance of patient autonomy in telerehabilitation identified by the experts in the present study. There are also significant differences between motor rehabilitation and psychotherapy in terms of TA. The home environment was seen by the experts as a challenge to ensure the patient's safety and sense of security. In telepsychology, on the other hand, the home environment is perceived as a safe place [72]. In (tele-)psychotherapy, the therapeutic dialogue represents the main therapeutic agent, whereas in physio- or occupational therapy, safeguarding the patient and therapeutic touch are relevant elements.

Themes and key aspects of TA in rehabilitation identified by our analysis confirmed those of existing (preliminary) frameworks of TA in physiotherapy, occupational therapy and speech and language therapy [16, 20, 21]. Additional aspects of TA in telerehabilitation as perceived by the experts included the support from others for ensuring physical safety, overall preparedness of the therapist and patient for telerehabilitation sessions and ensuring technical connectedness. Further aspects involved the necessity of relatives or caregivers applying touch after having received guidance i.e., acting as 'co-therapists', and promoting patient autonomy and motivation using additional digital options such as reminders, reward systems and display of therapy success. Considering these additional insights, our study contributes to a more comprehensive conceptualisation of TA in telerehabilitation. Apart from the TA between the patient and the therapist, it is important to acknowledge the impact of relatives or caregivers within both the physical and digital environment of the individual. Expanding the concept of TA may require an exploration of perspectives not only from therapists but also from patients and their family members. Furthermore, akin to physio-, occupational, or speech therapy [73], it seems crucial to address potential tensions or ruptures in the TA that may arise in telerehabilitation, requiring further attention in the development of an expanded concept of TA in this context.

## Reflexivity and limitations

Based on the professional background of mainly physiotherapy, the female study team may have influenced the results of this study. Various measures for achieving trustworthiness including credibility, dependability and transferability were undertaken in this study, however [74, 75]. Credibility was enhanced by choosing experts with various genders, ages and professional backgrounds, and different experiences in TA and telerehabilitation [74]. Semi-structured interviews and open-ended questions were used to gather rich information of the phenomenon under study. The type of qualitative content analysis was carefully selected based on the worldview of the researchers, research question and the approach to analysis, aiming to identify categories and subcategories, from which to derive central themes covering all relevant data [76]. To enhance dependability or trust, researchers were immersed in the research with their personal values and various backgrounds and used peer scrutiny and debriefing, identifying connections within the text [75]. To facilitate transferability, a thick description of the research team, worldviews, setting, timeline, data collection, and analysis was provided [75].

To ensure standardisation of the interviews, they were conducted on the basis of interview guides. A weakness in this respect is that the guides were not pilot tested before the interviews were conducted. This can be justified by the individuality of the different interview guides. Transcripts were not returned to participants for comment and/or correction and participants did not provide feedback on the findings, which may be another limitation of the study.

However, open questions were clarified during the interview. Given the small sample size, we may not have achieved saturation, which may be another limitation of this study [77]. Experts from various fields were interviewed, nonetheless, to gain insight into different perspectives on the research topic. Finally, we did not conduct interviews with patients to explore their views on establishing a trusting TA in telerehabilitation. Exploring the patient perspective on this matter is crucial for future research and the development of targeted interventions in this area.

## Conclusions

In the context of telerehabilitation, TA might influence the outcomes of treatment similar to conventional rehabilitation. Based on the literature search and expert interviews, it was possible to delineate and holistically map elements influencing successful TA for the setting of telerehabilitation. Three central themes were identified: building effective communication; nurturing a mutual relationship of trust and respect; and agreement on goals and tasks and drivers of motivation. In this qualitative study, key elements of TA in rehabilitation were expanded with additional elements of TA in telerehabilitation comprising support from caregivers for ensuring physical safety and technical connectedness, caregivers acting as co-therapists and applying professional touch, and promoting patient autonomy and motivation using specific strategies.

## Supporting information

**S1 File. Consolidated criteria for reporting qualitative studies (COREQ): 32-item checklist.**
(PDF)

**S1 Table. Research team characteristics and attributes.**
(PDF)

**S2 Table. Keywords used for the literature search and exemplary interview guide.**
(PDF)

**S3 Table. Influencing elements of a successful therapeutic alliance based on literature search.**
(PDF)

**S4 Table. Coding tree with descriptors.**
(PDF)

## Acknowledgments

We would like to thank all interviewed experts; without their willingness to provide information and commitment this work could not have been produced.

## Author Contributions

**Conceptualization:** Barbara Seebacher, Carole Geimer, Gudrun Diermayr.

**Data curation:** Barbara Seebacher, Carole Geimer, Maria Schwarz.

**Formal analysis:** Barbara Seebacher, Carole Geimer, Julia Neu, Gudrun Diermayr.

**Investigation:** Maria Schwarz.

**Methodology:** Barbara Seebacher, Maria Schwarz, Gudrun Diermayr.

**Project administration:** Barbara Seebacher.

**Software:** Carole Geimer, Julia Neu.

**Supervision:** Barbara Seebacher, Maria Schwarz, Gudrun Diermayr.

**Validation:** Maria Schwarz.

**Writing – original draft:** Barbara Seebacher.

**Writing – review & editing:** Barbara Seebacher, Carole Geimer, Julia Neu, Maria Schwarz, Gudrun Diermayr.

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
