## [Decision Letter · Decision Letter 0]

9 Jan 2024

PONE-D-23-09046Identifying central elements of the therapeutic alliance in the setting of telerehabilitation: a qualitative studyPLOS ONE

Dear Dr. Seebacher,

Thank you for submitting your manuscript to PLOS ONE. After careful consideration, we feel that it has merit but does not fully meet PLOS ONE’s publication criteria as it currently stands. Therefore, we invite you to submit a revised version of the manuscript that addresses the points raised during the review process.

Please submit your revised manuscript by Feb 23 2024 11:59PM. If you will need more time than this to complete your revisions, please reply to this message or contact the journal office at plosone@plos.org. Please include the following items when submitting your revised manuscript:A rebuttal letter that responds to each point raised by the academic editor and reviewer(s). You should upload this letter as a separate file labeled 'Response to Reviewers'.A marked-up copy of your manuscript that highlights changes made to the original version. You should upload this as a separate file labeled 'Revised Manuscript with Track Changes'.An unmarked version of your revised paper without tracked changes. You should upload this as a separate file labeled 'Manuscript'.

We look forward to receiving your revised manuscript.

Kind regards,

Nadinne Alexandra Roman, Ph.D.

Academic Editor

PLOS ONE

Journal Requirements:

3. We note that your Data Availability Statement is currently as follows: "All relevant data are within the manuscript and its Supporting Information files."

Reviewers' comments:

Reviewer's Responses to Questions

**Comments to the Author**

1. Is the manuscript technically sound, and do the data support the conclusions?

Reviewer #1: Yes

Reviewer #2: Partly

2. Has the statistical analysis been performed appropriately and rigorously? 

Reviewer #1: N/A

Reviewer #2: No

3. Have the authors made all data underlying the findings in their manuscript fully available?

Reviewer #1: Yes

Reviewer #2: Yes

4. Is the manuscript presented in an intelligible fashion and written in standard English?

Reviewer #1: Yes

Reviewer #2: Yes

5. Review Comments to the Author

Reviewer #1: Thank you for a timely and relevant paper for the readership of this journal. Telerehabilitation is an emerging area of practice and your paper provides important insights into the development of therapeutic alliance as part of this mode of service delivery. The Title adequately describes the study and alerts the reader to relevant content. The abstract is suitable for the nature of the paper, summarising relevant details in a measured and accurate manner. The introduction is clear and well structured, citing relevant literature, setting the scene for the study and alerting the reader to central concepts. The importance of therapeutic alliance is clearly outlined, however the link with telerehabilitation and why this requires specific attention is not strongly outlined, and instead is more implied. I would recommend the authors strengthen this in the introduction. The theoretical framework is described adequately and is appropriate for a qualitative study.

The ethical considerations are clearly described and align with expectations in the jurisdiction where the research was conducted, with due regard to participant confidentiality.

I am not convinced that the reporting of the literature review is adequate to support the readers understanding of your process in developing the interview guidelines. I would recommend that there is further narrative included in the paper on the nature of the issues identified in the literature review and the resultant interview guide areas should be described in broad terms.

The methods and procedures are clearly articulated and are appropriate for the study. The interview guides are described in the section on page 8, but there is not link to the literature to support the concepts included there. This needs to be strengthened.

The data analysis approach is transparent and clearly outlined, increasing veracity and rigor.

The results section begins with a brief paragraph on the literature review and points to the supplementary material in the main. There is inadequate narrative about the findings of the literature review as previously stated which requires attention.

The coding tree and themes section requires some revision to help the reader follow the results. On page 11, line 218, you use ethe term main concepts….but the table refers to codes. Using the term code consistently would help the reader. You also outline three central themes, but the way these are presented grammatically does not make it easy for the reader to understand what the themes are. I recommend you use semi colons to separate the main themes into 3 clear themes.

The themes are clearly described and supported by relevant and rich quotes to explain the concepts included in the theme.

You have developed a figure to visually represent the themes and the relationship to Therapeutic alliance. I am not convinced that the diagram represents the complexity of the relationship between the three themes, instead it simplifies them too much in to a linear, albeit circular relationship. The data does not support this. I recommend a reconsideration of the format of the figure, or potentially leaving it out.

The discussion is adequate but not well developed and could be further expanded to consider expansion of therapeutic alliance. You have identified relevant and appropriate limitations for the study, but have not considered the need to understand the patient perspective as an important area for future research.

The paper makes a great contribution to the field of telerehabilitation but requires further revisions to clarify the study reporting.

Reviewer #2: The title of the paper sounds very interesting for the audience, however, author need to address the following issues.

Introduction:

1. Therapeutic alliance required clear concise explanation for the audience. What does it compose of based on previous studies.

2. The hyphen inserted in the definition is placed in wrong way. See line 69.

3. The rational of the study with previous supporting literature is not clear.

Methodology:

1. The recruitment criteria of experts are not clear.

2. The inclusion criteria of the included papers are not clear.

3. How many papers were found and what number of the papers were included in study?

4. Is it possible to provide the questionnaires that were discussed with experts?

Result:

1. The steps of analysis are well written, however, the finding of the study and statistical analysis are not clear.

2. What were the suggestions and at what level of consent did the experts reached to identify the key elements of TA?

3. Check the Hyphen ( "..." ) insertion with respondence answers.

Discussion

1. The discussion is focused on three themes and looks good but it may need revision based on the result.

6. PLOS authors have the option to publish the peer review history of their article (what does this mean?). If published, this will include your full peer review and any attached files.

Reviewer #1: **Yes: **Dr Kim Bulkeley

Reviewer #2: **Yes: **Bishnu Dutta Acharya

---

## [Author Response · Author response to Decision Letter 0]

27 Jan 2024

PONE-D-23-09046

Identifying central elements of the therapeutic alliance in the setting of telerehabilitation: a qualitative study

PLOS ONE

Dear Academic Editor, Dear Dr Roman, 

On behalf of my co-authors, I would like to thank you for giving us the opportunity to submit a revised version of our manuscript.

We have revised the affiliations to meet PLOS ONE's style requirements.

We hereby confirm that our Data Availability Statement is correct as our manuscript and its Supporting Information files contain all raw data to replicate the results of our study. 

We endeavored to address Reviewer 2's comments as thoroughly as possible, even though they were not consistently clear to us. It is implied that the reviewer may not possess expertise in qualitative research based on the nature of the suggestions.

Please find our point-by-point responses to the Reviewers’ comments as follows.

Reviewers’ Comments to the Author

Reviewer #1: Thank you for a timely and relevant paper for the readership of this journal. Telerehabilitation is an emerging area of practice and your paper provides important insights into the development of therapeutic alliance as part of this mode of service delivery. The Title adequately describes the study and alerts the reader to relevant content. The abstract is suitable for the nature of the paper, summarising relevant details in a measured and accurate manner. The introduction is clear and well structured, citing relevant literature, setting the scene for the study and alerting the reader to central concepts. 

Dear Reviewer #1, dear Dr Bulkeley,

Thank you for reviewing our manuscript. We appreciate your valuable and constructive feedback. We have incorporated your feedback and hope that the presentation of both the process and findings of our qualitative study is now clearer. Please find our point-by-point response as follows.

Point 1: The importance of therapeutic alliance is clearly outlined, however the link with telerehabilitation and why this requires specific attention is not strongly outlined, and instead is more implied. I would recommend the authors strengthen this in the introduction. 

Response 1: Thank you for raising this point. We have added the following text to our introduction: 

‘TA in telerehabilitation requires special attention due to several key factors. Firstly, utilising technology-mediated communication in telerehabilitation associated with the potential disruptions from technical issues may affect the establishment and maintenance of the TA. Secondly, telerehabilitation may complicate deciphering non-verbal cues, which hold a significant role in face-to-face interactions and can contribute to the development of TA. Finally, the challenges of building trust and sustaining patient engagement may be heightened in telerehabilitation due to the constraints of limited physical presence.”

Point 2: The theoretical framework is described adequately and is appropriate for a qualitative study. The ethical considerations are clearly described and align with expectations in the jurisdiction where the research was conducted, with due regard to participant confidentiality. I am not convinced that the reporting of the literature review is adequate to support the readers understanding of your process in developing the interview guidelines. I would recommend that there is further narrative included in the paper on the nature of the issues identified in the literature review and the resultant interview guide areas should be described in broad terms.

Response 2: Thank you for this valuable advice. In the literature results section, we have now included a further narrative summarising the elements and issues identified in the literature review and resultant interview guide areas.

‘Relationship between the patient and therapist 

The essence of the TA lies in the intricate bond between the patient and therapist, encompassing personal, emotional, and professional dimensions. The therapist's crucial role involves displaying empathy and responsiveness, fostering mutual sharing of emotions, and engaging in self-revelation to build a strong personal relationship. [1]. Emphasising the importance of a common focus among all parties, the patient should be forefronted, transcending the focus from the illness to their individual physical, psychological, cultural, and social characteristics and needs in relation to personal goals and therapy contents [2-5]. Establishing a profound emotional and professional connection is essential for effective knowledge transmission at a professional level [6]. 

To maintain a client-centered approach, a clear distribution of roles and responsibilities is imperative to prevent conflict and encourage the active participation of the patient in the therapeutic process [7, 8]. Shared responsibility, identified as a key factor [9], necessitates a delicate balance between patient autonomy and therapist support, presenting a significant challenge [10]. Introducing humor serves to relegate the illness to the background, fostering an improved mood [1]. The therapist's congruence, involving genuine, open, and authentic interactions, is pivotal at a professional level [1]. The physical presence of the therapist is deemed relevant, especially in tele-rehabilitation, where ensuring safety [11], conducting a comprehensive physical examination [11, 12], and providing both psychological and physical support are challenging [9, 13]. Lack of physical presence emerges as a barrier to TA in tele-rehabilitation [14]. 

Trust 

Various aspects contribute to building mutual trust between the patient and therapist, including the exchange of information [2, 15], positive feedback, empathy and respect [2], along with sense of safety [11]. In tele-rehabilitation, the absence of direct safety guarantees poses a challenge to trust [16]. Information exchange is foundational for TA development [9], impacting interaction, satisfaction, and therapy success [17]. 

Communication

Effective communication enhances the rehabilitation process, enabling mutual understanding, shared decision-making, and person-centered communication [18]. Different communication channels, such as nonverbal, verbal, and paraverbal communication play crucial roles [19-21], with face-to-face communication being emphasised for nonverbal aspects [21]. Technical aspects in tele-rehabilitation[22] , including distance [23] and time-delay [24, 25], technical aspects [22], limit non-verbal communication, underscoring the importance of face-to-face interaction [24]. 

Agreement on goals and tasks

Agreement on goals between the therapist and patient is pivotal for goal setting, decision-making [7], TA development, and therapy implementation [19]. Goal setting, aligned with patient needs, increases motivation [26], while the patient's expectation of success, achievement of goals [6], autonomy, and self-management contribute to long-term therapy success [27]. Patient engagement becomes especially crucial in tele-rehabilitation [28-30]. 

External influencing factors

Relatives and/or caregivers form an integral part of the TA, often referred to as a triad [31]. Their support plays a significant role in promoting the rehabilitation process [1]. In cases involving children, a shared relationship is observed between the therapist, child, and parents [2]. Relatives and/or caregivers provide technical support during therapy preparation and procedure, ensuring safety [8] and the surroundings can influence TA development [14, 31]. Time is recognised as a valuable resource, influencing trust-building [32], communication, and the consideration of patient needs [33]. In tele-rehabilitation, spending more time with the patient is highlighted as a factor promoting TA [11].’

Point 3: The methods and procedures are clearly articulated and are appropriate for the study. The interview guides are described in the section on page 8, but there is not link to the literature to support the concepts included there. This needs to be strengthened.

Response 3: Acknowledging your justified suggestion, we have now included references supporting the concepts included in this paragraph.

‘In-depth semi-structured interviews included open questions about TA in motor rehabilitation in general [3, 9, 34], telerehabilitation and topics such as communication [22, 24], trust [4, 16], congruence [1], self-disclosure [1], respect [2], empathy [1, 35], therapist-patient relationship [1, 2], working alliance (incl. goal setting, tasks, motivation) [6-8], external influencing factors (e.g., relatives and/or caregivers, information and communication technology, safety, time) [14, 31, 33], roles and responsibilities [2, 8].”

Point 4: The data analysis approach is transparent and clearly outlined, increasing veracity and rigor. The results section begins with a brief paragraph on the literature review and points to the supplementary material in the main. There is inadequate narrative about the findings of the literature review as previously stated which requires attention.

Response 4: Kindly review the modifications made to the manuscript and our response to point 2, addressing the previously mentioned shortcomings in the narrative regarding the literature review findings.

Point 5: The coding tree and themes section requires some revision to help the reader follow the results. On page 11, line 218, you use ethe term main concepts….but the table refers to codes. Using the term code consistently would help the reader. You also outline three central themes, but the way these are presented grammatically does not make it easy for the reader to understand what the themes are. I recommend you use semi colons to separate the main themes into 3 clear themes. The themes are clearly described and supported by relevant and rich quotes to explain the concepts included in the theme.

Response 5: We appreciate your valuable feedback. The section has been revised accordingly, and we have provided further clarification on the theme numbers. The last sentence has been added in response to a comment from Reviewer 2.

‘We identified 9 unique codes and 50 subcodes as potential elements related to TA through qualitative content analysis in our interview data (Table 2; see S4 Table for the full coding tree including code descriptors). The intercoder reliability for the raters was determined to be Kappa = 0.900 (p<0.0001), 95% CI (0.873, 0.927). Further synthesis generated three central themes: building effective communication (theme 1); nurturing a mutual relationship of trust and respect (theme 2); agreement on goals and tasks and drivers of motivation (theme 3). Representative quotes are presented with the themes.’

Point 6: You have developed a figure to visually represent the themes and the relationship to Therapeutic alliance. I am not convinced that the diagram represents the complexity of the relationship between the three themes, instead it simplifies them too much in to a linear, albeit circular relationship. The data does not support this. I recommend a reconsideration of the format of the figure, or potentially leaving it out.

Response 6: We acknowledge your perspective and have explored various formats for the figure. However, none of them appeared to effectively communicate the intricate relationship among the three themes. Consequently, we have chosen to exclude the figure.

Point 7: The discussion is adequate but not well developed and could be further expanded to consider expansion of therapeutic alliance. 

Response 7: Thank you for your guidance. We have incorporated the following paragraph into the discussion section:

‘Considering these additional insights, our study contributes to a more comprehensive conceptualisation of TA in telerehabilitation. Apart from the TA between the patient and the therapist, it is important to acknowledge the impact of relatives or caregivers within both the physical and digital environment of the individual. Expanding the concept of TA may require an exploration of perspectives not only from therapists but also from patients and their family members. Furthermore, akin to physio-, occupational, or speech therapy [36], it seems crucial to address potential tensions or ruptures in the TA that may arise in telerehabilitation, requiring further attention in the development of an expanded concept of TA in this context.”

Point 8: You have identified relevant and appropriate limitations for the study but have not considered the need to understand the patient perspective as an important area for future research. The paper makes a great contribution to the field of telerehabilitation but requires further revisions to clarify the study reporting.

Response 8: Thank you for raising this important point. We have now included the following study limitation:

‘Finally, we did not conduct interviews with patients to explore their views on establishing a trusting TA in telerehabilitation. Exploring the patient perspective on this matter is crucial for future research and the development of targeted interventions in this area.”

Reviewer #2: The title of the paper sounds very interesting for the audience, however, author need to address the following issues.

Dear Reviewer #2, dear Mr Acharya,

Thank you for evaluating our manuscript and offering your suggestions to enhance its quality. Please find our point-by-point response as follows.

Introduction:

Point 1: Therapeutic alliance required clear concise explanation for the audience. What does it compose of based on previous studies.

Response 1: Thank you for raising this point. We have defined and described TA using previous literature on pages 1 and 2. From your suggestion and the comment from Reviewer 1, we acknowledge however, that we missed to clearly outline the link with telerehabilitation and why this requires specific attention. We have therefore now added the following text to our introduction: 

‘TA in telerehabilitation requires special attention due to several key factors. Firstly, utilising technology-mediated communication in telerehabilitation associated with the potential disruptions from technical issues may affect the establishment and maintenance of the TA. Secondly, telerehabilitation may complicate deciphering non-verbal cues, which hold a significant role in face-to-face interactions and can contribute to the development of TA. Finally, the challenges of building trust and sustaining patient engagement may be heightened in telerehabilitation due to the constraints of limited physical presence.”

Point 2: The hyphen inserted in the definition is placed in wrong way. See line 69.

Response 2: In this sentence, there is no hyphen; however, you may refer to the quotation marks. We appreciate your advice and have adjusted the type and placement of the quotation marks as follows: 

Therapeutic alliance (TA), which originated in psychotherapy [5], has been defined as ‘a mutual collaboration and partnership between the therapist and client’ [6, page 1].

Point 3: The rational of the study with previous supporting literature is not clear.

Response 3: Thank you for this comment. As mentioned in Point 1, we acknowledge that we missed clearly outlining the rationale of the study. Therefore, we have now added the following text to our introduction: 

‘TA in telerehabilitation requires special attention due to several key factors. Firstly, utilising technology-mediated communication in telerehabilitation associated with the potential disruptions from technical issues may affect the establishment and maintenance of the TA. Secondly, telerehabilitation may complicate deciphering non-verbal cues, which hold a significant role in face-to-face interactions and can contribute to the development of TA. Finally, the challenges of building trust and sustaining patient engagement may be heightened in telerehabilitation due to the constraints of limited physical presence.”

Methodology:

Point 4: The recruitment criteria of experts are not clear.

Response 4: In a qualitative study, researchers may opt for a relatively small and purposively selected sample [37], focusing on enhancing depth rather than breadth of understanding [38]. Purposive sampling is employed to select respondents likely to provide relevant and valuable information [39] and is a strategy for efficiently utilising limited research resources by identifying and selecting pertinent cases [38].

Purposive sampling strategies diverge from random sampling methods, aiming to ensure that particular types of cases, those potentially relevant to the research study, are included in the final sample. The decision to adopt a purposive strategy is grounded in the belief that, aligning with the study's goals and objectives, individuals of certain characteristics may offer distinct and valuable perspectives on the ideas and issues under consideration, justifying their inclusion in the sample [40, 41]. Quota sampling offers increased flexibility, specifying categories and setting a minimum number for each, rather than mandating fixed numbers of cases with specific criteria [40].

We recognise our omission in specifying the minimum number of experts required for each quota. To address this, we have included the pertinent details on quota sampling and the minimum number of experts for each group in the recruitment criteria. The revised sentence is as follows:

‘These were categorised into the following groups, each requiring a minimum of 2 experts, and purposive quota sampling [40] was performed.”

Point 5: The inclusion criteria of the included papers are not clear.

Response 5: We appreciate your highlighting this crucial aspect. The inclusion criteria for the included papers have been incorporated into the relevant section of the manuscript:

‘Aims of the literature review were to complement Bordin's theoretical construct of TA within rehabilitation and telerehabilitation, particularly within the domains of physiotherapy, occupational therapy, or speech and language therapy. Peer-reviewed full-text articles, books, and conference proceedings in English or German language were included, that investigated relevant aspects of TA within rehabilitation or telerehabilitation and identified gaps in the literature, explored new perspectives, or proposed enhancements to existing TA frameworks.”

Point 6: How many papers were found and what number of the papers were included in study?

Response 6: We appreciate your valuable feedback. In accordance with your suggestion, we have incorporated the requested information into our results: 

‘A total of 7,933 articles were initially identified after removing duplicates. Upon screening titles and abstracts, 7,821 studies were excluded. Subsequently, 182 potentially relevant papers underwent full-text evaluation. Following the full-text screening process, 145 studies were excluded for the following reasons: the evaluation solely focused on the effects of interventions on TA (n=38); exclusive focus on the perceptions of patients or therapists regarding TA (n=15); utilisation of Bordin’s theoretical construct of TA without extension (n=14); investigation of TA in the domain of psychotherapy or psychology (n=57); or exploration of TA in the field of telemedicine or digital healthcare (n=23). Finally, 35 articles were included.”

Point 7: Is it possible to provide the questionnaires that were discussed with experts?

Response 7: Certainly, this is possible. The questions for the experts have already been supplied in S2 Table, titled ‘Keywords used for the literature search and exemplary interview guide.

Results:

Point 8: The steps of analysis are well written, however, the finding of the study and statistical analysis are not clear.

Response 8: Thank you for bringing up this concern. In the subsequent paragraphs, we aim to elucidate our approach, and we hope that this will offer ample clarification regarding the findings of our study and the statistical analysis.

This qualitative study employed a combined deductive and inductive approach to inquiry. Data collection centred on interviews, with a focus on delving deeply into the perspectives and experiences of experts. Participant selection was purposeful, aligning with the study's objectives and emphasising depth over breadth, resulting in smaller sample sizes compared to quantitative studies [37]. The data analysis process utilised inductive reasoning, allowing themes and patterns to emerge organically. A Steigleder modified variant of a structuring qualitative content analysis, a common qualitative analysis method, was employed [42]. Qualitative studies prioritise contextual understanding, exploring the meaning and interpretation of experiences within specific social, cultural, or organisational contexts. Given our study's aims, statistical analysis was deemed inappropriate, with the exception of calculating Cohen’s kappa. For further clarification and details, please refer to our explanation provided in Point 9.

Findings from qualitative studies often unveil recurring themes or patterns in participants' narratives. Our study provides detailed and rich descriptions of experts' perspectives and experiences [43]. Following qualitative research principles, we incorporated quotes to illustrate key points and offer insights into the contextual factors influencing the studied phenomenon [37]. Furthermore, our qualitative inquiry may contribute to a more comprehensive conceptualisation of TA in telerehabilitation, particularly within physiotherapy, occupational therapy, and speech and language therapy. Acknowledging the subjectivity of researchers, our findings include reflections on how our background, beliefs, or biases may have influenced the study [43]. In conclusion, we are confident that our qualitative study offers a holistic and nuanced understanding of the central elements of TA in the context of telerehabilitation by exploring the depth and context of experts' experiences and perspectives.

If we should include any of this or any further information in our manuscript, we are more than willing to do so. 

Point 9: What were the suggestions and at what level of consent did the experts reached to identify the key elements of TA?

Response 9: While the exact intent of the question is not entirely clear, we aim to provide a comprehensive response. Steigleder’s modified variant of a structuring qualitative content analysis [42] does not incorporate reliability analysis, such as Kappa statistics, to determine agreement between interviewees. The suggestions provided by the experts are extensively detailed in the results section of the manuscript and in S4 Table. 

Upon your insightful inquiry, we have now included information on our intercoder agreement evaluation utilising Cohen’s kappa statistics in our manuscript:

Ensuring reliability is crucial in content analysis [44] because it guarantees consistency in coding decisions and assesses the rigour and transparency of the coding frame and its application to the data [45, 46]. 

To assess intercoder reliability using GraphPad Prism 9, San Diego, California, we calculated Cohen’s kappa along with its 95% confidence interval (CI) [47, 48]. McHugh recommends interpreting kappa values as follows: 0-0.20 indicating no agreement, 0.21-0.39 minimum agreement, 0.40-0.59 weak agreement, 0.60-0.79 moderate agreement, 0.80-0.90 strong agreement, and 0.91-1 almost perfect agreement [49]. 

The intercoder reliability for the raters was determined to be Kappa = 0.900 (p<0.0001), 95% CI (0.873, 0.927).

Point 10: Check the Hyphen ( "..." ) insertion with respondence answers.

Response 10: Thank you, we have edited all quotation marks.

Discussion

Point 11: The discussion is focused on three themes and looks good but it may need revision based on the result.

Response 11: Thank you for your input; we have integrated all the relevant modifications into the results within the discussion section.

References

1. Bishop M, Kayes N, McPherson K. Understanding the therapeutic alliance in stroke rehabilitation. Disability and Rehabilitation. 2021;43(8):1074-83. doi: 10.1080/09638288.2019.1651909.

2. Crom A, Paap D, Wijma A, Dijkstra PU, Pool G. Between the Lines: A Qualitative Phenomenological Analysis of the Therapeutic Alliance in Pediatric Physical Therapy. Physical & Occupational Therapy In Pediatrics. 2020;40(1):1-14. doi: 10.1080/01942638.2019.1610138.

3. Moore AJ, Holden MA, Foster NE, Jinks C. Therapeutic alliance facilitates adherence to physiotherapy-led exercise and physical activity for older adults with knee pain: a longitudinal qualitative study. J Physiother. 2020;66(1):45-53. Epub 2019/12/18. doi: 10.1016/j.jphys.2019.11.004. PubMed PMID: 31843425.

4. Lawford BJ, Bennell KL, Campbell PK, Kasza J, Hinman RS. Therapeutic Alliance Between Physical Therapists and Patients With Knee Osteoarthritis Consulting Via Telephone: A Longitudinal Study. Arthritis Care & Research. 2020;72(5):652-60. doi: https://doi.org/10.1002/acr.23890.

5. Gómez Penedo JM, Babl AM, Grosse Holtforth M, Hohagen F, Krieger T, Lutz W, et al. The Association of Therapeutic Alliance With Long-Term Outcome in a Guided Internet Intervention for Depression: Secondary Analysis From a Randomized Control Trial. J Med Internet Res. 2020;22(3):e15824. Epub 2020/03/25. doi: 10.2196/15824. PubMed PMID: 32207689; PubMed Central PMCID: PMCPMC7139432.

6. Vestøl I, Debesay J, Pajalic Z, Bergland A. The importance of a good therapeutic alliance in promoting exercise motivation in a group of older Norwegians in the subacute phase of hip fracture; a qualitative study. BMC Geriatrics. 2020;20(1):118. doi: 10.1186/s12877-020-01518-7.

7. Ranner M, Guidetti S, von Koch L, Tham K. Experiences of participating in a client-centred ADL intervention after stroke. Disability and Rehabilitation. 2019;41(25):3025-33. doi: 10.1080/09638288.2018.1483434.

8. Wilms IL. The computerized cognitive training alliance - A proposal for a therapeutic alliance model for home-based computerized cognitive training. Heliyon. 2020;6(1):e03254. Epub 2020/02/12. doi: 10.1016/j.heliyon.2020.e03254. PubMed PMID: 32042977; PubMed Central PMCID: PMCPMC7002830.

9. Miciak M, Mayan M, Brown C, Joyce AS, Gross DP. A framework for establishing connections in physiotherapy practice. Physiotherapy Theory and Practice. 2019;35(1):40-56. doi: 10.1080/09593985.2018.1434707.

10. Cott C. Client-centred rehabilitation: client perspectives. Disability and Rehabilitation. 2004;26(24):1411-22. doi: 10.1080/09638280400000237.

11. Shulver W, Killington M, Morris C, Crotty M. ‘Well, if the kids can do it, I can do it’: older rehabilitation patients' experiences of telerehabilitation. Health Expectations. 2017;20(1):120-9. doi: https://doi.org/10.1111/hex.12443.

12. Kairy D, Tousignant M, Leclerc N, Côté A-M, Levasseur M, Researchers TT. The patient's perspective of in-home telerehabilitation physiotherapy services following total knee arthroplasty. International journal of environmental research and public health. 2013;10(9):3998-4011. doi: 10.3390/ijerph10093998. PubMed PMID: 23999548.

13. Miciak M, Mayan M, Brown C, Joyce AS, Gross DP. The necessary conditions of engagement for the therapeutic relationship in physiotherapy: an interpretive description study. Archives of Physiotherapy. 2018;8(1):3. doi: 10.1186/s40945-018-0044-1.

14. Cranen K, Drossaert CHC, Brinkman ES, Braakman‐Jansen ALM, Ijzerman MJ, Vollenbroek‐Hutten MMR. An exploration of chronic pain patients’ perceptions of home telerehabilitation services. Health Expectations: An International Journal of Public Participation in Health Care & Health Policy. 2012;15(4):339-50. doi: 10.1111/j.1369-7625.2011.00668.x.

15. Price B. Developing patient rapport, trust and therapeutic relationships. Nurs Stand. 2017;31(50):52-63. Epub 2017/08/10. doi: 10.7748/ns.2017.e10909. PubMed PMID: 28792344.

16. Held JP, Ferrer B, Mainetti R, Steblin A, Hertler B, Moreno-Conde A, et al. Autonomous rehabilitation at stroke patients home for balance and gait: safety, usability and compliance of a virtual reality system. European Journal of Physical and Rehabilitation Medicine 2018;54(4):545-53. Epub 2017/09/28. doi: 10.23736/s1973-9087.17.04802-x. PubMed PMID: 28949120.

17. O'Keeffe M, Cullinane P, Hurley J, Leahy I, Bunzli S, O'Sullivan PB, et al. What Influences Patient-Therapist Interactions in Musculoskeletal Physical Therapy? Qualitative Systematic Review and Meta-Synthesis. Physical Therapy. 2016;96(5):609-22. doi: 10.2522/ptj.20150240.

18. Jesus TS, Silva IL. Toward an evidence-based patient-provider communication in rehabilitation: linking communication elements to better rehabilitation outcomes. Clinical Rehabilitation. 2015;30(4):315-28. doi: 10.1177/0269215515585133.

19. Babatunde F, MacDermid J, MacIntyre N. Characteristics of therapeutic alliance in musculoskeletal physiotherapy and occupational therapy practice: a scoping review of the literature. BMC health services research. 2017;17(1):375. doi: 10.1186/s12913-017-2311-3.

20. Fuchs T. Non-verbale Kommunikation: Phänomenologische, ent­wicklungspsy­chologi­sche und therapeutische Aspekte. Zeitschrift für Klinische Psychologie und Psychotherapie. 2003;51:333-45.

21. Geuter U. Praxis Körperpsychotherapie – 10 Prinzipien der Arbeit im therapeutischen Prozess. Berlin, Deutschland: Springer; 2019.

22. Wade SL, Raj SP, Moscato EL, Narad ME. Clinician perspectives delivering telehealth interventions to children/families impacted by pediatric traumatic brain injury. Rehabilitation Psychology. 2019;64(3):298-306. doi: 10.1037/rep0000268.

23. Egolf DB. Nonverbal Communication and Telerehabilitation. In: Kumar S, Cohn ER, editors. Telerehabilitation. Health Informatics. London: Springer; 2013. p. 41–54.

24. Lie SS, Karlsen B, Graue M, Oftedal B. The influence of an eHealth intervention for adults with type 2 diabetes on the patient–nurse relationship: a qualitative study. Scandinavian Journal of Caring Sciences. 2019;33(3):741-9. doi: https://doi.org/10.1111/scs.12671.

25. Oftedal B, Kolltveit B-CH, Graue M, Zoffmann V, Karlsen B, Thorne S, et al. Reconfiguring clinical communication in the electronic counselling context: The nuances of disruption. Nursing Open. 2019;6(2):393-400. doi: https://doi.org/10.1002/nop2.218.

26. Oyake K, Suzuki M, Otaka Y, Tanaka S. Motivational Strategies for Stroke Rehabilitation: A Descriptive Cross-Sectional Study. Frontiers in Neurology. 2020;11. doi: 10.3389/fneur.2020.00553.

27. Dobkin BH. Behavioral self-management strategies for practice and exercise should be included in neurologic rehabilitation trials and care. Current opinion in neurology. 2016;29(6):693-9. doi: 10.1097/wco.0000000000000380. PubMed PMID: 00019052-201612000-00005.

28. Dinesen B, Nielsen G, Andreasen JJ, Spindler H. Integration of Rehabilitation Activities Into Everyday Life Through Telerehabilitation: Qualitative Study of Cardiac Patients and Their Partners. Journal of medical Internet research. 2019;21(4):e13281. Epub 2019/04/16. doi: 10.2196/13281. PubMed PMID: 30985284; PubMed Central PMCID: PMCPMC6487348.

29. Graffigna G, Barello S, Bonanomi A, Menichetti J. The Motivating Function of Healthcare Professional in eHealth and mHealth Interventions for Type 2 Diabetes Patients and the Mediating Role of Patient Engagement. Journal of Diabetes Research. 2016;2016:2974521. doi: 10.1155/2016/2974521.

30. Grünloh C, Myreteg G, Cajander Å, Rexhepi H. "Why Do They Need to Check Me?" Patient Participation Through eHealth and the Doctor-Patient Relationship: Qualitative Study. Journal of medical Internet research. 2018;20(1):e11. Epub 2018/01/18. doi: 10.2196/jmir.8444. PubMed PMID: 29335237; PubMed Central PMCID: PMCPMC5789160.

31. Lawton M, Sage K, Haddock G, Conroy P, Serrant L. Speech and language therapists’ perspectives of therapeutic alliance construction and maintenance in aphasia rehabilitation post-stroke. International Journal of Language & Communication Disorders. 2018;53(3):550-63. doi: 10.1111/1460-6984.12368.

32. Wilson S, Chaloner N, Osborn M, Gauntlett-Gilbert J. Psychologically informed physiotherapy for chronic pain: patient experiences of treatment and therapeutic process. Physiotherapy. 2017;103(1):98-105. doi: 10.1016/j.physio.2015.11.005.

33. Gard G. Factors important for good interaction in physiotherapy treatment of persons who have undergone torture: A qualitative study. Physiotherapy Theory and Practice. 2007;23(1):47-55. doi: 10.1080/09593980701209584.

34. Miciak M, Mayan M, Brown C, Joyce AS, Gross DP. The necessary conditions of engagement for the therapeutic relationship in physiotherapy: an interpretive description study. Arch Physiother. 2018;8:3. Epub 2018/02/23. doi: 10.1186/s40945-018-0044-1. PubMed PMID: 29468089; PubMed Central PMCID: PMCPMC5816533.

35. Oyake K, Suzuki M, Otaka Y, Tanaka S. Motivational Strategies for Stroke Rehabilitation: A Descriptive Cross-Sectional Study. Front Neurol. 2020;11:553. Epub 2020/06/27. doi: 10.3389/fneur.2020.00553. PubMed PMID: 32587572; PubMed Central PMCID: PMCPMC7297944.

36. Miciak M, Rossettini G. Looking at Both Sides of the Coin: Addressing Rupture of the Therapeutic Relationship in Musculoskeletal Physical Therapy/Physiotherapy. J Orthop Sports Phys Ther. 2022;52(8):500-4. Epub 2022/06/21. doi: 10.2519/jospt.2022.11152. PubMed PMID: 35722761.

37. Miles MB, Huberman AM, Saldana J. Qualitative Data Analysis: A Methods Sourcebook. 4th ed. Los Angeles, London, New Delhi, Singapore, Washington DC Sage Publications; 2019.

38. Palinkas LA, Horwitz SM, Green CA, Wisdom JP, Duan N, Hoagwood K. Purposeful Sampling for Qualitative Data Collection and Analysis in Mixed Method Implementation Research. Adm Policy Ment Health. 2015;42(5):533-44. Epub 2013/11/07. doi: 10.1007/s10488-013-0528-y. PubMed PMID: 24193818; PubMed Central PMCID: PMCPMC4012002.

39. Kelly S. Qualitative interviewing techniques and styles. In: Bourgeault I, Dingwall R, de Vries R, editors. The Sage Handbook of Qualitative Methods in Health Research. Thousand Oaks: Sage Publications; 2010.

40. Mason J. Qualitative researching. London: Sage Publications Ltd; 2017.

41. Robinson OC. Sampling in Interview-Based Qualitative Research: A Theoretical and Practical Guide. Qualitative Research in Psychology. 2014;11(1):25-41. doi: 10.1080/14780887.2013.801543.

42. Steigleder S. Die strukturierende qualitative Inhaltsanalyse im Praxistest: eine konstruktiv kritische Studie zur Auswertungsmethodik von Philipp Mayring: Tectum-Verlag; 2008.

43. Patton MQ. Qualitative Research & Evaluation Methods: Integrating Theory and Practice. 4th ed. Thousand Oaks, CA: Sage Publications, Inc.; 2015.

44. Neuendorf KA. The Content Analysis Guidebook. Thousand Oaks, California2017. Available from: https://methods.sagepub.com/book/the-content-analysis-guidebook-2e.

45. Hruschka DJ, Schwartz D, St.John DC, Picone-Decaro E, Jenkins RA, Carey JW. Reliability in Coding Open-Ended Data: Lessons Learned from HIV Behavioral Research. Field Methods. 2004;16(3):307-31. doi: 10.1177/1525822X04266540.

46. MacPhail C, Khoza N, Abler L, Ranganathan M. Process guidelines for establishing Intercoder Reliability in qualitative studies. Qualitative Research. 2015;16(2):198-212. doi: 10.1177/1468794115577012.

47. Brennan RL, Prediger DJ. Coefficient Kappa: Some Uses, Misuses, and Alternatives. Educational and Psychological Measurement. 1981;41(3):687-99. doi: 10.1177/001316448104100307.

48. Cohen J. A Coefficient of Agreement for Nominal Scales. Educational and Psychological Measurement. 1960;20(1):37-46. doi: 10.1177/001316446002000104.

49. McHugh ML. Interrater reliability: the kappa statistic. Biochemia medica. 2012;22(3):276-82. Epub 2012/10/25. PubMed PMID: 23092060; PubMed Central PMCID: PMCPMC3900052.

---

## [Decision Letter · Decision Letter 1]

19 Feb 2024

Identifying central elements of the therapeutic alliance in the setting of telerehabilitation: a qualitative study

PONE-D-23-09046R1

Dear Dr. Seebacher,

We’re pleased to inform you that your manuscript has been judged scientifically suitable for publication and will be formally accepted for publication once it meets all outstanding technical requirements.

Kind regards,

Nadinne Alexandra Roman, Ph.D.

Academic Editor

PLOS ONE

Additional Editor Comments (optional):

Reviewers' comments:

Reviewer's Responses to Questions

**Comments to the Author**

1. If the authors have adequately addressed your comments raised in a previous round of review and you feel that this manuscript is now acceptable for publication, you may indicate that here to bypass the “Comments to the Author” section, enter your conflict of interest statement in the “Confidential to Editor” section, and submit your "Accept" recommendation.

Reviewer #1: All comments have been addressed

Reviewer #2: All comments have been addressed

2. Is the manuscript technically sound, and do the data support the conclusions?

Reviewer #1: (No Response)

Reviewer #2: Yes

3. Has the statistical analysis been performed appropriately and rigorously? 

Reviewer #1: (No Response)

Reviewer #2: Yes

4. Have the authors made all data underlying the findings in their manuscript fully available?

Reviewer #1: (No Response)

Reviewer #2: Yes

5. Is the manuscript presented in an intelligible fashion and written in standard English?

Reviewer #1: (No Response)

Reviewer #2: Yes

6. Review Comments to the Author

Reviewer #1: Thank you for your thoughtful and comprehensive response to the reviewers comments. This paper makes an important contribution to the telepractice discourse around therapeutic alliance, which is an emerging area of research.

Reviewer #2: Thank you for addressing the comments of reviewers. Different professional will find this paper interesting. Correct the line 581 physio- (Delete -, after physio).

7. PLOS authors have the option to publish the peer review history of their article (what does this mean?). If published, this will include your full peer review and any attached files.

Reviewer #1: **Yes: **Dr Kim Bulkeley

Reviewer #2: **Yes: **Bishnu Dutta Acharya

---

## [Editor Report · Acceptance letter]

23 Feb 2024

PONE-D-23-09046R1 

PLOS ONE

Dear Dr. Seebacher, 

I'm pleased to inform you that your manuscript has been deemed suitable for publication in PLOS ONE. Congratulations! Your manuscript is now being handed over to our production team.

Kind regards, 

on behalf of

Dr. Nadinne Alexandra Roman 

Academic Editor

PLOS ONE